

# Exploring the plasmodesmata callose-binding protein gene family in upland cotton: unraveling insights for enhancing fiber length

Haibo Zhang[1,2], Xianghui Xiao[2], Ziyin Li[2], Yu Chen[2], Pengtao Li[2], Renhai Peng[2], Quanwei Lu[1,2] and Youwu Wang[1]

[1] College of Agriculture, Tarim University, Alar, China
[2] School of Biotechnology and Food Engineering, Anyang Institute of Technology, Anyang, Henan, China

## ABSTRACT

Plasmodesmata are transmembrane channels embedded within the cell wall that can facilitate the intercellular communication in plants. Plasmodesmata callose-binding (PDCB) protein that associates with the plasmodesmata contributes to cell wall extension. Given that the elongation of cotton fiber cells correlates with the dynamics of the cell wall, this protein can be related to the cotton fiber elongation. This study sought to identify PDCB family members within the *Gossypium. hirsutum* genome and to elucidate their expression profiles. A total of 45 distinct family members were observed through the identification and screening processes. The analysis of their physicochemical properties revealed the similarity in the amino acid composition and molecular weight across most members. The phylogenetic analysis facilitated the construction of an evolutionary tree, categorizing these members into five groups mainly distributed on 20 chromosomes. The fine mapping results facilitated a tissue-specific examination of group V, revealing that the expression level of *GhPDCB9* peaked five days after flowering. The VIGS experiments resulted in a marked decrease in the gene expression level and a significant reduction in the mature fiber length, averaging a shortening of 1.43–4.77 mm. The results indicated that *GhPDCB9* played a pivotal role in the cotton fiber development and served as a candidate for enhancing cotton yield.

# INTRODUCTION

Cell walls are essential for plant development and growth. They are a crucial and dynamic structural component that provides vital mechanical support throughout plant growth, development, and adaptation to varying environmental conditions. The cell wall consists of a complex yet well-organized extracellular matrix that encompasses aromatic compounds, including polysaccharides, glycoproteins, and lignin. Notable components of the cell wall matrix include corpus callosum, cellulose, hemicellulose, and pectin (*Jamet & Dunand, 2020*).

Corresponding authors
Quanwei Lu, daweianyang@163.com
Youwu Wang, wangyw1975@126.com

Plasmodesmata callose-binding protein (PDCB) is a class of proteins identified in cell walls that affects plasmodesma transport. The C-terminus of the PDCB protein contains an X8 domain with a signal peptide at the N-terminus. The X8 protein domain is prevalent among various *Arabidopsis* proteins and serves as a non-enzymatic accessory domain for some β-1,3 glucanases. Callose, which is primarily composed of a β-1,3-linked glucose homopolymer, is a major component of the cell wall. β-1,3 glucanases are involved in plasmodesmata permeability (*Zavaliev et al., 2013*). Plasmodesmata (PD) is a transmembrane passage traversing the cell wall, establishing a community continuum throughout most of the plant. PDs serve as transport channels for various signaling molecules, including small and large biomolecules, and are crucial for the growth, development, and defense mechanisms of higher plants. The initial study of plasmodesmata in intercellular substance transport within plant leaves originated in tobacco leaves, where over 90% of PDs were simple in young or mature leaves and functioned as photosynthetic reservoirs (*Oparka et al., 1999*). *Lee et al. (2005)* conducted experiments in which anchored cell wall protein fragments were extracted from tobacco suspension culture cells using TMV-MP as a phosphorylated substrate. They identified a 35 Da protein-PAPK with calcium-dependent protein kinase activity. Callose deposition was observed on the cell plate during segmentation and in the neck areas of mature and immature tissues surrounding PDs. This process is integral to symplastic segregation during the elongation of cotton fibers and pollen maturation (*Lee et al., 2005*). *Simpson et al. (2009)* discovered a new protein gene that was related to PD, *AT5G61130*, which is regulated by either the CaMV 35S promoter or its native promoter. When fused with the YFP coding sequence and temporarily expressed in tobacco protoplasts, epidermal cell walls exhibited fluorescent spots. In cavernous mesophyll cells, these fluorescent spots were restricted to interwall junctions among the surrounding cells and maintained cell wall separation. These results suggested that this gene is associated with the cell wall (*Simpson et al., 2009*).

Cotton, a crucial commercial crop, is used as a raw material in the textile industry. Globally, there are approximately 45 diploid cotton species and seven tetraploid cotton species (*Grover et al., 2015*). Advances in genome sequencing technology have enabled the complete sequencing of various cotton genomes, including TM-1 (*G. hirsutum L.*) (*Li et al., 2015*), *G. arboreum* (*Li et al., 2014*), and *G. raimondii* (*Wang et al., 2012*). A comprehensive and user-friendly repository comprising 11,226 QTLs and significant marker-trait associations derived from 136 studies related to various fiber quality characteristics in crops was provided (*Gudi et al., 2024*). However, there are currently no reported analyses of the PDCB gene family at the whole-genome level in cotton. This study systematically identified the PDCB family gene and conducted bioinformatic analysis, offering insights into its number, subcellular localization, chromosomal distribution, evolutionary relationships, and motif analysis. *GH_A12G1651* (*GhPDCB9*) was identified as a potential contributor to fiber length and its role was validated using VIGS in cotton. A comprehensive examination of PDCB on a genome-wide scale can provide a critical foundation for enhancing cotton production through molecular breeding.

## MATERIALS AND METHODS

### Identification of PDCB genes in upland cotton

The amino acid and genome sequences of the *Arabidopsis* PDCB gene family were obtained from the website of TAIR (http://www.arabidopsis.org/), whereas the amino acid and CDS sequences of the TM-1 (*G. hirsutum L.* second generation genome) genome were obtained from the College of Agriculture and Biotechnology, Zhejiang University (http://cotton.zju.edu.cn/). To analyze the PDCB gene family of *Arabidopsis thaliana* and upland cotton (*G. hirsutum*), the exploitation tool of TBtools was employed. The *A. thaliana* sequence was used as the query sequence for the BLAST search of the cotton genome database (*Chen et al., 2020*). To predict the physicochemical properties of PDCB proteins, including the instability index, isoelectric point (PI), molecular weight (MW), and amino acid count, the website-based ProtParam tool was used (https://web.expasy.org/protparam/). We predicted the subcellular localization of PDCBs using CELLO v.2.5 resources (http://cello.life.nctu.edu.tw/) and WoLF PSORT (https://wolfpsort.hgc.jp/) (*Yu et al., 2006*).

### Evolutionary analysis of PDCB gene family in cotton

The amino acid sequences of the PDCB gene family members in *Arabidopsis thaliana* and upland cotton were analyzed and compared, leading to the construction of a phylogenetic tree. MEGA-X was employed to construct a neighbor-joining tree for PDCB genes, with 1,000 bootstrap replications (*Hall, 2013*). The resulting phylogenetic tree was visualized using the EvolView software (*He et al., 2016*).

### Gene structure and conserved motifs analyses

Cotton PDCB members were analyzed using the MEME online software (http://meme-suite.org/). Amino acid sequences were inputted, and the software detected the number and types of motifs. The analysis was conducted using the following parameters: a maximum of 10 motifs was displayed, while other settings remained at their default values. Gene and conserved motif structures, including exons and introns, were visualized using TBTools.

### Chromosome location analysis of PDCB gene family

The positions of the members contained in the PDCB gene family on the chromosome were extracted and organized using TBtools according to those of the annotated genes in the gff3 file. Subsequently, the physical distribution of these PDCB members was visualized using the MapInspect software.

### Analysis of the PDCB cis-regulatory element

To investigate the PDCB promoter in *G .hirsutum* and to predict the function of the PDCB gene, the analysis focused on regions predominantly 1,500 bp upstream of the gene, with a few exceptions. For thoroughness, additional fragments extending 500 bp upstream were analyzed. Therefore, a 2,000 bp sequence upstream from the initial codon of the gene was extracted for detailed examination. This sequence was then subjected to the analysis using Plantcare (https://bioinformatics.psb.ugent.be/webtools/plantcare/html/).

## Quantitative real-time polymerase chain reaction analysis

At the full flowering stage, fiber samples from *G. hirsutum L.* (cv. CCRI45) cultivated in the field were collected at intervals of 5 days post-anthesis (DPA), 10 DPA, 15 DPA and 20 DPA, along with tissue samples from the stems(tender stem), leaves(fresh leaves the size of fingernails), and flowers. Three different biological replicates were taken for each sample and three technical replicates were performed. These samples were immediately frozen in liquid nitrogen and preserved at −80 °C to facilitate the subsequent extraction of total RNA.

The RNA Prep Pure Plant Kit (DP441; Tiangen, Beijing, China) was used to extract total RNA from each sample, and RNA quality was assessed using a Nanodrop2000 nucleic acid analyzer and gel electrophoresis. Subsequently, TranScript All-in-One First-Strand cDNA Synthesis SuperMix for qPCR (TransGen Biotech, Beijing, China) was used to synthesize cDNAs. An ABI 7500 Fast Real-Time PCR system (Applied Biosystems, Foster City, CA, USA) was used to perform RT-qPCR based on the TransStart Top Green qPCR SuperMix kit protocol (Transgen Biotech, Beijing, China). Primer-BLAST, sourced from the online NCBI database, was used to design specific primers for the differentially expressed genes (DEGs), with details provided in Table S1. Using the primer sequences R: 5′-TGTCCGTCAGGCAACTCAT-3′ and F: 5′-ATCCTCCGTCTTGACCTTG-3′, the housekeeping β-actin gene was used as a reference to standardize relative expression levels. Relative gene expression levels were quantified using the $2^{-\Delta\Delta Ct}$ method (*Livak & Schmittgen, 2001*).

## Virus induced gene silencing of the candidate genes

The procedure began by selecting a 300 bp viral encoding fragment that exhibited the best alignment with *Gh_A12G1651* (https://vigs.solgenomics.net/). Subsequently, the fragments were amplified using the specific primers, virus induced gene silencing (VIGS): *Gh_A12G1651*-R and VIGS: *Gh_A12G1651*-F. The amplified fragment was ligated to the Clcrv vector after double digestion with *Spe*I and *Asc*I. After sequencing, correct plasmid transformation was performed using *Agrobacterium* (LBA4404). It was then introduced into TM-1 through leaf back injection. Upon the emergence of albino plants, DNA extraction was verified in positive seedlings. Subsequently, RNA was reverse-transcribed for RT-qPCR analysis to assess expression levels. Phenotype identification was performed once the fibers reached maturity.

# RESULTS

## Identification of *PDCB* gene family members of *G. hirsutum* and analysis of their basic physical and chemical properties

The identified genes were screened for specific domains. A total of 45 members of the PDCB gene family were successfully identified in *G. hirsutum* (TM-1). Following the standard protein naming convention, these 45 members of the TM-1 genome were designated *GhPDCB1–GhPDCB45*. The fundamental physicochemical characteristics were predicted and analyzed (Table 1).

**Table 1** Information on the PDCB gene family in the genome of *Gossypium hirsutum* L. genome.

| Gene name | Sequence ID | Number of amino acid | M W | pI | Instability index | Aliphatic index | Subcellular prediction |
|---|---|---|---|---|---|---|---|
| *GhPDCB1* | *Gh_A12G0700* | 415 | 45,237.14 | 5.87 | 50.94 | 84.58 | E.R. |
| *GhPDCB2* | *Gh_D12G0713* | 424 | 46,328.50 | 6.07 | 52.18 | 85.07 | E.R. |
| *GhPDCB3* | *Gh_A07G1664* | 291 | 31,472.65 | 8.15 | 74.11 | 60.03 | nuclear |
| *GhPDCB4* | *Gh_D07G1873* | 291 | 31,428.58 | 8.19 | 71.48 | 60.03 | extra |
| *GhPDCB5* | *Gh_A12G0427* | 176 | 18,816.45 | 4.96 | 32.60 | 81.53 | extra |
| *GhPDCB6* | *Gh_A10G1112* | 118 | 12,461.05 | 5.32 | 64.70 | 69.75 | extra |
| *GhPDCB7* | *Gh_D02G0284* | 198 | 20,945.81 | 7.57 | 58.45 | 80.40 | golg_plas |
| *GhPDCB8* | *Gh_A02G0218* | 198 | 20,884.79 | 7.57 | 57.66 | 79.90 | golg_plas |
| *GhPDCB9* | *Gh_A12G1651* | 209 | 21,015.26 | 7.54 | 24.06 | 53.73 | extra |
| *GhPDCB10* | *Gh_A03G0620* | 198 | 20,154.51 | 5.99 | 28.47 | 67.07 | extracellular |
| *GhPDCB11* | *Gh_D03G0905* | 198 | 20,164.54 | 5.99 | 26.30 | 67.07 | extra |
| *GhPDCB12* | *Gh_A10G1095* | 300 | 32,394.46 | 6.57 | 67.74 | 74.13 | extra |
| *GhPDCB13* | *Gh_D12G1221* | 160 | 17,770.54 | 8.98 | 29.84 | 82.38 | plas |
| *GhPDCB14* | *Gh_D02G0295* | 258 | 27,475.46 | 5.37 | 73.19 | 64.30 | nuclear |
| *GhPDCB15* | *Gh_D10G1416* | 291 | 31,358.87 | 5.92 | 65.03 | 66.70 | chloroplast |
| *GhPDCB16* | *Gh_A02G0230* | 265 | 28,102.16 | 5.88 | 67.58 | 61.13 | nuclear |
| *GhPDCB17* | *Gh_A07G2272* | 245 | 26,259.58 | 8.09 | 54.00 | 51.76 | chloroplast |
| *GhPDCB18* | *Gh_D10G1394* | 207 | 21,575.36 | 4.47 | 51.65 | 76.52 | extra |
| *GhPDCB19* | *Gh_A05G1268* | 278 | 28,630.10 | 4.59 | 54.40 | 64.24 | plas |
| *GhPDCB20* | *Gh_D11G2836* | 362 | 37,282.18 | 8.08 | 48.63 | 74.59 | chloroplast |
| *GhPDCB21* | *Gh_D10G1104* | 175 | 18,804.04 | 5.59 | 39.78 | 76.46 | extra |
| *GhPDCB22* | *Gh_D12G1801* | 210 | 21,042.28 | 7.54 | 25.12 | 53.05 | extra |
| *GhPDCB23* | *Gh_A10G1366* | 175 | 1,873.00 | 5.68 | 45.01 | 76.46 | chloroplast |
| *GhPDCB24* | *Gh_D07G0822* | 245 | 26,285.70 | 7.65 | 54.93 | 55.35 | chloroplast |
| *GhPDCB25* | *Gh_D06G1708* | 112 | 12,414.24 | 6.38 | 30.80 | 87.23 | extra |
| *GhPDCB26* | *Gh_A05G3280* | 242 | 25,467.82 | 8.99 | 51.70 | 66.53 | extra |
| *GhPDCB27* | *Gh_D07G1666* | 116 | 13,087.03 | 5.25 | 38.30 | 74.05 | extra |
| *GhPDCB28* | *Gh_A10G1967* | 125 | 13,475.27 | 6.80 | 31.15 | 68.72 | extra |
| *GhPDCB29* | *Gh_A07G1521* | 127 | 14,398.56 | 6.81 | 44.31 | 69.13 | extra |
| *GhPDCB30* | *Gh_A09G0754* | 328 | 33,972.97 | 5.70 | 38.07 | 46.43 | chloroplast |
| *GhPDCB31* | *Gh_D04G0327* | 236 | 24,659.74 | 8.12 | 49.31 | 62.46 | chloroplast |
| *GhPDCB32* | *Gh_D02G1230* | 332 | 35,746.88 | 5.53 | 70.53 | 60.48 | nuclear |
| *GhPDCB33* | *Gh_D05G1437* | 278 | 28,677.16 | 4.77 | 53.25 | 64.60 | plas |
| *GhPDCB34* | *Gh_D05G2949* | 177 | 18,133.24 | 4.51 | 39.79 | 66.78 | extra |
| *GhPDCB35* | *Gh_Sca016465G01* | 115 | 12,524.57 | 8.51 | 27.72 | 82.35 | extra |
| *GhPDCB38* | *Gh_D13G0849* | 188 | 19,321.65 | 5.99 | 32.94 | 61.86 | extra |
| *GhPDCB39* | *Gh_A13G0725* | 188 | 19,251.64 | 5.98 | 29.91 | 63.40 | extra |

**Table 1** (*continued*)

| Gene name | Sequence ID | Number of amino acid | M W | pI | Instability index | Aliphatic index | Subcellular prediction |
|-----------|-------------|----------------------|------|------|-------------------|-----------------|------------------------|
| *GhPDCB40* | *Gh_A06G2003* | 129 | 13,875.88 | 5.81 | 34.81 | 81.78 | chloroplast |
| *GhPDCB41* | *Gh_A06G1366* | 118 | 13,175.99 | 6.02 | 32.42 | 78.64 | extra |
| *GhPDCB42* | *Gh_A10G1966* | 134 | 14,583.33 | 5.80 | 44.92 | 59.03 | extra |
| *GhPDCB43* | *Gh_D10G2247* | 148 | 16,257.16 | 6.56 | 34.91 | 62.03 | extra |
| *GhPDCB44* | *Gh_D12G0424* | 97 | 10,545.80 | 6.52 | 38.45 | 67.53 | chloroplast |
| *GhPDCB45* | *Gh_A01G0299* | 56 | 6,337.23 | 7.65 | 30.60 | 61.07 | extra |

**Notes.**
pI, isoelectric point; MW, molecular weight (mass).

The analysis indicated that the length of the amino acid sequences within the gene family varied, ranging from 112 amino sequences within the gene family varied, ranging from 112 amino acids in *GhPDCB25* to 424 amino acids in *GhPDCB2*. However, the sequences of *GhPDCB44* and *GhPDCB45* were shorter than 100 amino acids, consisting of 97 and 56 amino acids, respectively. The isoelectric points of these proteins spanned from 4.47 (*GhPDCB18*) to 8.99 (*GhPDCB26*). The instability index indicated protein stability in the test tube ($\leq 40$, possibly stable; $>40$, possibly unstable). Furthermore, subcellular localization predictions indicated that 24 proteins were localized extracellularly, nine within the chloroplast, and four within the nucleus.

## Phylogenetic tree analysis of the cotton PDCB gene family

This study involved multiple sequence alignments of *Arabidopsis* and cotton members, followed by the construction of a phylogenetic tree constructed from the PDCB amino acid sequences. The results presented a classification pattern similar to that of *A. thaliana*, in which the cotton PDCB gene family members were categorized into five distinct groups (I–IV) (Fig. 1). Specifically, in Group I, cotton exhibited seven members and *A. thaliana* had three members. In Group II, cotton comprised 11 members and *A. thaliana* comprised 15 members. Group III had the fewest members, with cotton possessing six members, and *Arabidopsis* containing one. In Group IV, cotton had eight members, and *A. thaliana* had two. In Group V, cotton encompassed 13 members, and *A. thaliana* had eight members.

## Analysis of PDCB gene structure and protein conserved motifs in *G. hirsutum*

Analysis of protein motifs using MEME identified 20 potential motifs (Fig. 2A). Motif 1 was included in all *GhPDCB* proteins, indicating that it is conserved within the *GhPDCB* family. Notably, *GhPDCB6*, *GhPDCB7*, *GhPDCB8*, *GhPDCB9*, *GhPDCB10*, *GhPDCB11*, *GhPDCB18*, *GhPDCB19*, *GhPDCB20*, *GhPDCB22*, *GhPDCB33*, *GhPDCB34*, *GhPDCB36*, *GhPDCB38*, and *GhPDCB39* exhibited similarities with motifs 4–7, with motif 15 appearing as the first. In the third group, *GhPDCB1* and *GhPDCB2* shared motif 8 and held the same positions. Similarly, within the fifth group, *GhPDCB9*, *GhPDCB10*, *GhPDCB11*, *GhPDCB38*, and *GhPDCB39* displayed an exact match for motif 7, in a consistent order. The distribution of protein domains within the PDCB family (Fig. 2B) revealed that eight *GhPDCB*s possessed domains beyond the common X8 and Glyco_hydro superfamily

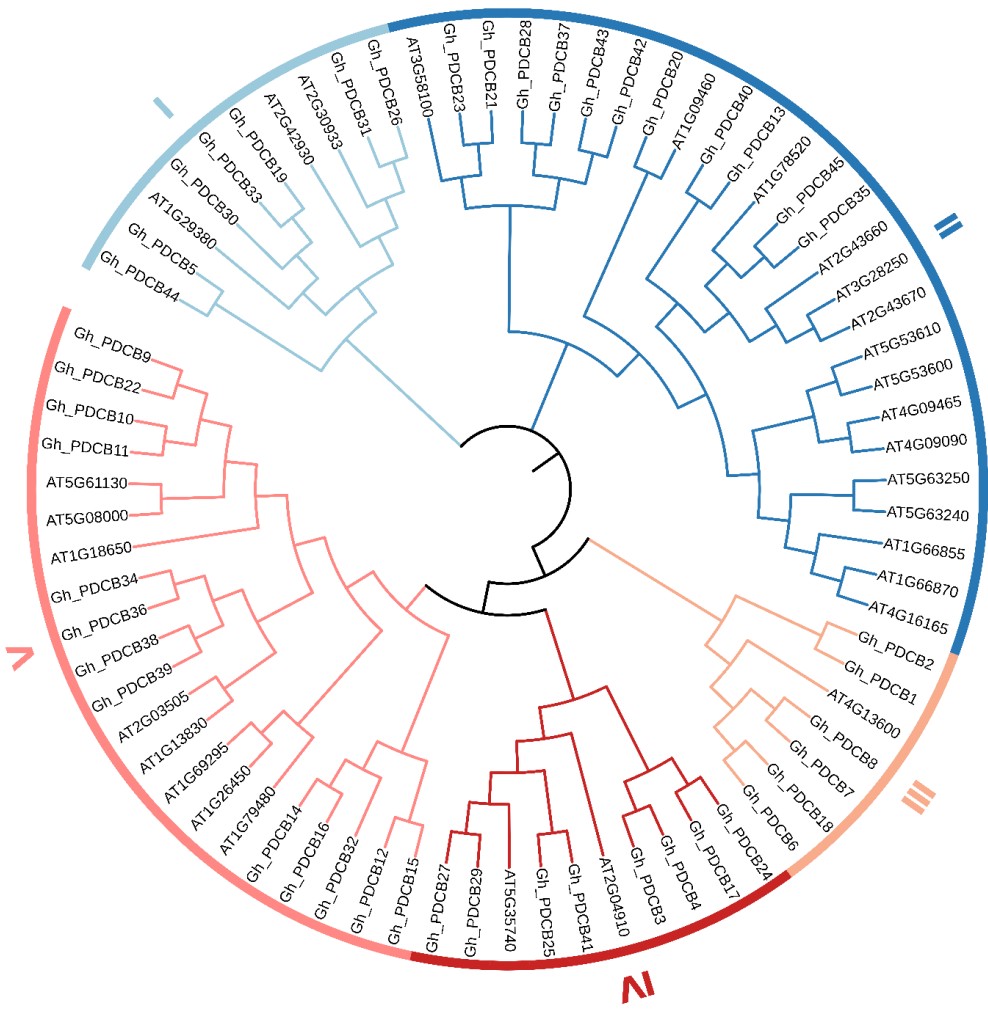

**Figure 1 Phylogenetic tree of the PDCB gene family in upland cotton.**

domains. Further analysis of gene structure (Fig. 2C) highlighted similarities in the number of introns and exons among most family members, indicating their close evolutionary relationships. Notably, the majority of the PDCB genes contained 3–4 introns, whereas a few had 1–2 introns.

## Analysis of chromosome mapping of PDCB family genes

Based on physical mapping of PDCB family members, 42 out of the 45 PDCB genes were situated on 20 chromosomes, with the remaining three genes positioned on a scaffold (Fig. 3). Physical mapping of the PDCB gene family revealed that 42 out of 45 genes were distributed across 20 chromosomes, and the remaining three genes were situated on scaffolds (Fig. 3). Notably, there were variations in the gene count and chromosomal locations between the At and Dt subgenomes. For instance, the chromosomes A04 and A11 lacked genes, whereas the chromosomes D04 and D11 possessed one gene as well.

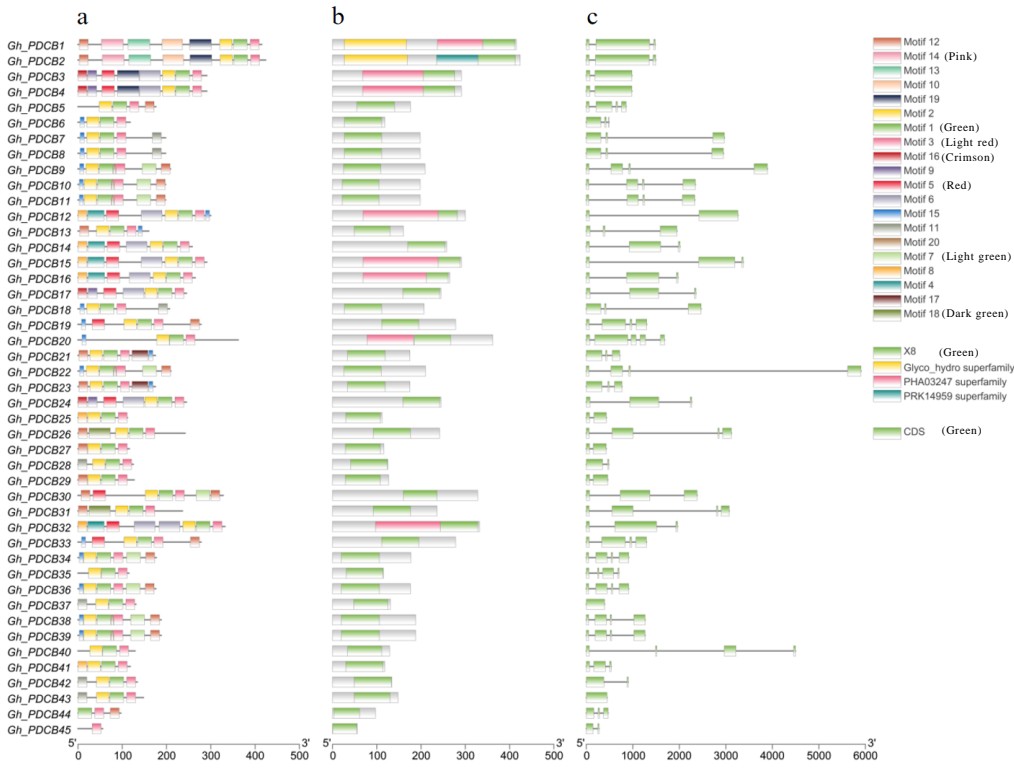

**Figure 2** **Exon-intron structure and conserved motif of PDCB genes in *G. hirsutum*.** (A) Analysis of conserved motif of GhPDCB protein sequences. Different motifs are shown in a specific color. (B) Gh-PDCB Protein domain prediction. (C) Intron and exon analysis of *GhPDCB* genes.

Conversely, chromosomes A05 and A12 each contained three genes, chromosome D05 had two genes, and D12 was characterized by four genes.

## Collinearity analysis of the cotton PDCB gene family

To further investigate the PDCB gene family in upland cotton, a collinearity analysis was conducted *G. hirsutum* and with *G. arboretum*, *G. barbadense* and *A. thaliana* (Fig. 4). The analysis revealed homologous regions across the At and Dt subgenomes of upland cotton, demonstrating significant collinearity within species (Fig. 4A). Interspecific comparisons identified 31 instances of collinearity between *G. hirsutum* and *G. arboreum* (Fig. 4B), 57 instances between *G. barbadense* and *G. hirsutum* (Fig. 4C), and 19 instances between *G. hirsutum* and *A. thaliana* (Fig. 4D). This analysis shows that there is a strong collinearity within *G. hirsutum* and *G. arboretum*, *G. barbadense* and *A. thaliana*, indicating that they are significantly conservative in evolution.

### *GhPDCB* gene promoter analysis

The analysis of the 2,000 bp upstream promoter region of the PDCB gene, conducted *via* PlantCare, revealed the presence of various cis-regulatory elements, as depicted in Fig. 5. The hormonal regulation played a pivotal role in the development of cotton fibers. This analysis highlighted more cis-regulatory elements associated with hormone

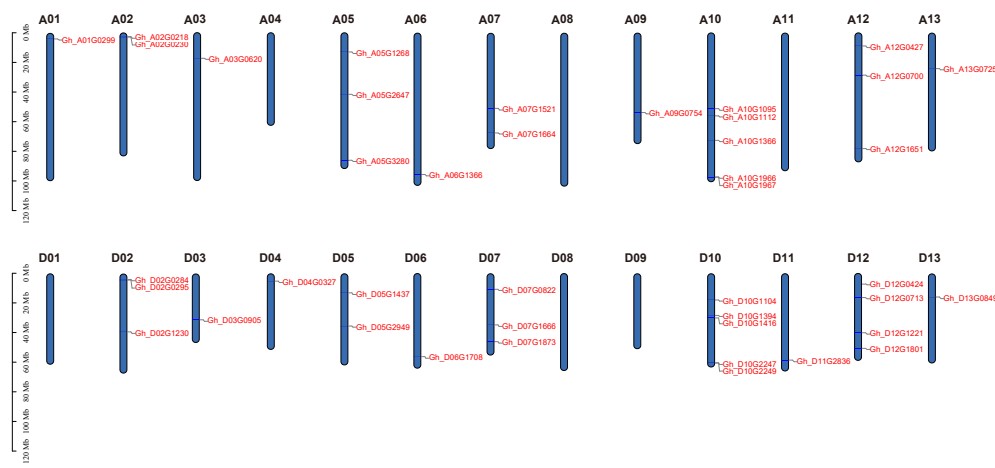

**Figure 3 Location of the PDCB gene on the chromosome.** Chromosome names are shown on above and gene names are shown on the right.

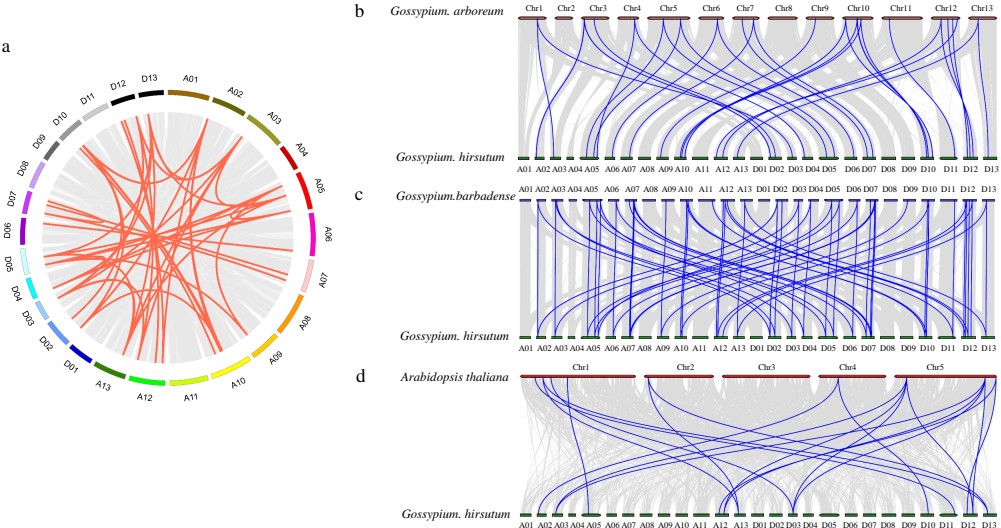

**Figure 4 Collinearity analysis of PDCB gene family in cotton.** (A) Synteny analysis among *G hirsutum* (At subgenome) and *G. hirsutum* (Dt subgenome). (B) Synteny analysis among *G. arboreum*, and *G. hirsutum*. (C) Synteny analysis among *G. barbadense* and *G. hirsutum*. (D) Synteny analysis among *G. hirsutum* and *A. thaliana*.

responses, including those responsive to salicylic acid, gibberellin, auxin, and abscisic acid, suggesting that these genes may influence fiber elongation by modulating hormonal response processes. Additionally, certain cis-regulatory elements related to abiotic stresses were identified, covering aspects such as light reactions, anaerobic induction, defense, stress

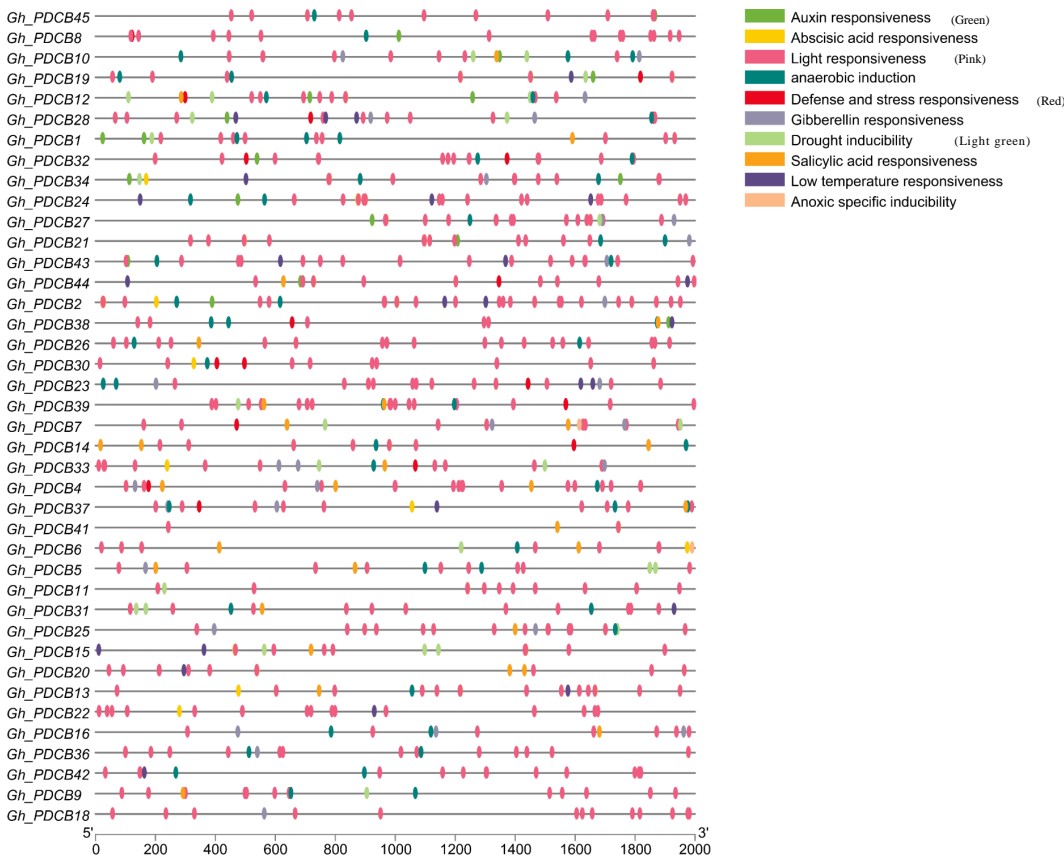

**Figure 5** **Analysis of *cis*-regulatory elements in promoters of *GhPDCB* genes.** Cis-regulatory elements with the same function are shown in the same color.

responsiveness, and drought inducibility, with light response elements being particularly prevalent and broadly distributed.

## Analysis of expression patterns

Based on the results of fine mapping (PDCB-3, *GH_A12G2014*, *GhPDCB9*, *QFL-A12-5*) and the constructed evolutionary tree, genes homologous to *GhPDCB9* were selected for the quantitative fluorescence analysis across three distinct tissue types (stem, leaf, and flower) and four developmental stages of fiber (5, 10, 15, and 20 DPA) (Figs. 6A and 6B).

RT-qPCR analysis of these 12 genes revealed that they shared similar expression patterns. These genes exhibited broad tissue expression profiles. *GhPDCB9*, *GhPDCB22*, *GhPDCB10*, *GhPDCB11*, and *GhPDCB38* displayed notably higher expression levels in the stem, surpassing those observed in other tissues. This increased expression suggests their potential involvement in cell elongation. Conversely, *GhPDCB36* was predominantly expressed in flowers, thus regulating flower development.

*GhPDCB9* and *GhPDCB22* expression levels exhibited a consistent reduction from 5 to 15 DPA, suggesting a potential association with fiber elongation. In this gene group, *GhPDCB10*, *GhPDCB11*, *GhPDCB34*, *GhPDCB38*, and *GhPDCB12* displayed a progressive

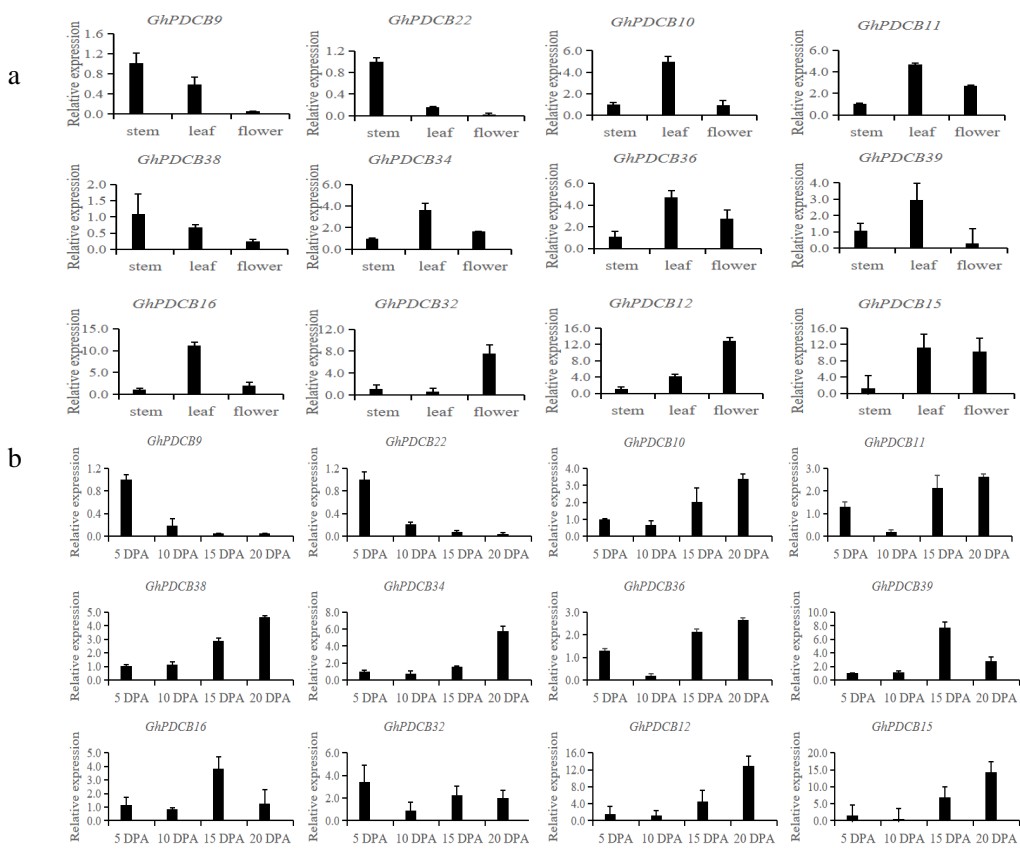

**Figure 6** **Analysis of the expression patterns in different tissues.** (A) Relative transcript abundance of each gene in the different tissues. (B) Transcript abundance of different genes during different flowering periods.

increase in expression levels from 5 to 20 DPA, primarily during late fiber development, indicating their involvement in secondary wall biosynthesis. These findings indicate a specific relationship between these genes and fiber development.

## VIGS of *GhPDCB9* in TM-1

The RT-qPCR results indicated that *GhPDCB9* could affect the development of cotton fiber at 5 DPA, which was consistent with findings from previous QTL research (*Lu et al., 2021*). Consequently, the next phase involved a detailed examination of *GhPDCB9*'s role. The function of *GhPDCB9* was investigated using virus-induced gene silencing. In this experimental setup, PDS encodes a gene for chlorophyll synthesis. As a result of PDS gene silencing, cotton leaves failed to synthesize chlorophyll, with bleaching of new leaves at later stages. This bleaching served as a positive control for successful silencing. The albino phenotype was observed following infection with the cotton strain, confirming effective gene silencing (Fig. 7A). Subsequently, the DNA extracted from the gene-silenced cotton leaves was subjected to specific primer testing, enabling the selection of correctly silenced cotton plants for fluorescence quantification. The expression levels of *GhPDCB9*

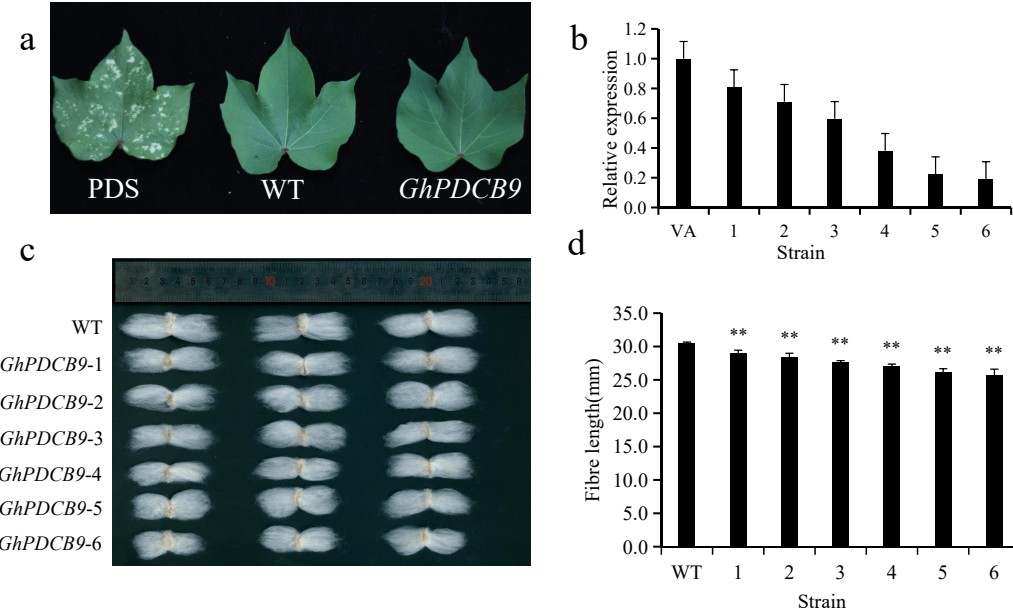

**Figure 7  Silencing of *GhPDCB9* using VIGS.** (A) The PDS whitening control. (B) Silencing efficiency of the silenced plants. (C) Mature fibers of the silenced plants. (D) Statistical analysis of average mature fibre length of WT andtransgenic lines. Values represent mean ± SD, **, $P < 0.01$.

were verified using RT-qPCR. Notably, the silenced plants exhibited a significant decrease in *GhPDCB9* gene expression levels after silencing (Fig. 7B). RT-qPCR analysis revealed that the cotton fibers demonstrated enhanced silencing efficiency. A descriptive statistical method was used to analyze the length of mature fibers. The fiber length experienced a notable reduction after virus-induced silencing compared to that of the control group, with an average shortening ranging from 1.43 to 4.77 mm. These differences were statistically significant (Figs. 7C and 7D).

# DISCUSSION

Cotton fiber, a vital natural resource in the textile industry, is derived from the epidermal cells of the ovule. Among cotton varieties, upland cotton is the main species for cultivation because of its superior fiber yield and robust adaptability, encompassing approximately 95% of the total cotton cultivation area (*Yoo & Wendel, 2014*). Cotton fiber development comprises four sequential phases: initiation, cell elongation, secondary wall thickening, and maturation, wherein the length of the epidermal cells extends from to 10–20 μm to 3–6 cm. During the initial two phases, the fiber number and length were predominantly affected, whereas the later two stages were associated with fiber strength and thickness (*Patel et al., 2020*). The development and growth of cotton fibers are influenced by several factors, including transcription factors, pH levels, and plant hormones, such as IAA, GA, and brassinolide (*Beasley, 1973*; *Raghavendra, Ye & Kinoshita, 2023*; *Sun et al., 2019*; *Suo et al., 2003*). Because *A. thaliana* trichomes have a developmental pattern resembling that of

cotton fibers, the genes promoting trichome advancement in *A. thaliana* are pivotal for initiating cotton fiber development (*Li et al., 2023*).

The plasmodesmata callose protein family is extensive, featuring the X8 domain with a signal sequence that enables the attachment of glycosylphosphatidylinositol to the external surface of the plasma membrane. The X8 domain was distinguished by its persistent configuration of one Phe residue and six Cys residues (Fig. 2), indicating its significance in carbohydrate binding (*Barral et al., 2005*). In *Arabidopsis*, a significant connection between intercellular communication and PDCB-mediated callose deposition has been established. Although this gene family has been extensively investigated in *Arabidopsis*, research on cotton is rare. Recent studies have identified a noteworthy association between SSR markers HAU0734 and *qFL-12-5* and the physical position of *G. hirsutum L.,* which is closely linked to *GhUGT103*. Gene expression patterns also indicate a relationship with fiber development (*Xiao et al., 2019*). Using high-density genetic maps, two QTL associated with fiber strength and six QTL related to fiber length were identified across the four chromosomes. Through integration of transcriptome data obtained from both qPCR analysis and parental lines, four genes linked to the QTL were identified. Among these, plasmodesmata callose-binding protein 3 (PDCB-3, *GH_A12G2014*, *GhPDCB9*, of *qFL-A12-5*) has emerged as a promising candidate gene with implications for fiber length (*Lu et al., 2021*).

A total of 45 PDCB genes were selected from upland cotton based on the unique X8 domain associated with PDCB. They share similar conserved protein motifs, and typically consist of two exons. Further investigation revealed that each pair of linear homologous genes exhibited identical or similar subcellular localization and sequences, suggesting parallel evolutionary homologs with comparable gene functions. The analysis of the 2,000 bp sequence upstream of the promoter revealed that cis-regulatory elements contained both hormone-related and abiotic stress-related elements. Given that hormones exerted a regulatory impact on fiber development, *GhPDCB9* could regulate the fiber development through its involvement in the hormone synthesis. Furthermore, the prevalence of cis-regulatory elements associated with abiotic stress demonstrated a potential relationship with plant stress resistance, which may require further research. Notably, QTL mapping identified *qFL-A12-5*, which was linked to fiber length, prompting the selection of 12 genes from the *GhPDCB9* family for expression pattern analysis. RT-qPCR analysis revealed that the 12 genes displayed similar expression patterns. Among these, *GhPDCB9*, *GhPDCB22*, *GhPDCB10*, *GhPDCB11*, and *GhPDCB38* exhibited notably higher expression levels in the stem than in the other tissues. *GhPDCB36* was predominantly expressed in flowers, indicating its role in the regulation of flower development. Subsequently, gene silencing experiments targeting *GhPDCB9* (VIGS) led to a significant reduction in fiber length, with an average shortening ranging from 1.43 to 4.77 mm. These findings suggest that *GhPDCB9* is associated with fiber elongation and development. Overall, the observed characterizations indicate the significance of *GhPDCB* in cotton, with the identified gene *GhPDCB9* (*GH_A12G2014*) serving as a valuable genetic resource for enhancing cotton fiber yield.

## CONCLUSIONS

In this study, 45 PDCB genes were identified in upland cotton and were categorized into five groups. Subsequently, an analysis of the PDCB gene characteristics in cotton was conducted, encompassing physical and chemical properties, phylogenetic relationships, conserved motifs, gene structures, and evolutionary collinearity. Our findings demonstrate the evolutionary conservation of PDCB genes across various ploid cotton species. RT-qPCR analysis was performed to confirm the significant *GhPDCB9* expression during fiber elongation. In the VIGS experiment, we observed a significant reduction in cotton fiber length following gene silencing, compared to that in the control group. The results indicated that *GhPDCB9* was related to fiber elongation and required further investigation. Our study sheds light on the potential contributions of PDCB genes to cotton production, offering valuable insights into strategies for improving cotton fiber length.

### Funding
This study was funded by the National Natural Science Foundation of China (32272188,32272179, 32070560). The funders had no role in study design, data collection and analysis, decision to publish, or preparation of the manuscript.

### Grant Disclosures
The following grant information was disclosed by the authors:
The National Natural Science Foundation of China: 32272188, 32272179, 32070560.

### Competing Interests
The authors declare there are no competing interests.

### Author Contributions
- Haibo Zhang conceived and designed the experiments, performed the experiments, analyzed the data, prepared figures and/or tables, authored or reviewed drafts of the article, and approved the final draft.
- Xianghui Xiao conceived and designed the experiments, performed the experiments, analyzed the data, prepared figures and/or tables, authored or reviewed drafts of the article, and approved the final draft.
- Ziyin Li performed the experiments, analyzed the data, prepared figures and/or tables, and approved the final draft.
- Yu Chen performed the experiments, analyzed the data, prepared figures and/or tables, and approved the final draft.
- Pengtao Li analyzed the data, authored or reviewed drafts of the article, and approved the final draft.
- Renhai Peng analyzed the data, authored or reviewed drafts of the article, and approved the final draft.
- Quanwei Lu conceived and designed the experiments, analyzed the data, authored or reviewed drafts of the article, and approved the final draft.

- Youwu Wang conceived and designed the experiments, analyzed the data, authored or reviewed drafts of the article, and approved the final draft.

## Data Availability

The raw measurements are available in the Supplementary Files.

## Supplemental Information

Supplemental information for this article can be found online at http://dx.doi.org/10.7717/peerj.17625#supplemental-information.

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
