# Peer review of "Exploring the plasmodesmata callose-binding protein gene family in upland cotton: unraveling insights for enhancing fiber length"

_PeerJ, doi:10.7717/peerj.17625_

## Round 0.1 · original submission · Major Revisions

Dear Authors

The manuscript cannot be accepted for publication in its current form. It needs a major revision before publication. The authors are invited to revise the paper considering all the suggestions made by the reviewers. Please note that the requested changes are required for publication.

With Thanks

**Language Note:** The review process has identified that the English language must be improved. PeerJ can provide language editing services - please contact us at [email protected] for pricing (be sure to provide your manuscript number and title). Alternatively, you should make your own arrangements to improve the language quality and provide details in your response letter. – PeerJ Staff

·

Basic reporting

Lines 37, 152, and 158: Arabidopsis needs to be italicized.
Line 56: Replace “them” with “the”.

Experimental design

Figure 7:
a. Please detail the statistical analysis that was used.
b. In Figure 7d, strain 3 is depicted as different from the rest, however, there is no visually significant difference. It would be ideal to double-check the results obtained before publishing these data.

Validity of the findings

No comments

·

Basic reporting

I suggest to cite the most interesting recent article published in cotton quality traits.
• Gudi S*, Pavan M, Alagappan P, Raigar OP, Halladakeri P, Gowda RSR, Kumar P, Singh G, Tamta M, Susmitha S, Amandeep, and Saini DK (2024) Fashion meets science: how advanced breeding approaches could revolutionize the textile industry. Crit. Rev. Biotechnol. doi: 10.1080/07388551.2024.2314309

Include botanical name in title
L19 use full name of genus for first time use
L27 Keywords need to be changed. At least five keywords must be present. These must cove all topic and should not include the words used in title. It helps in searching the article more easily.
Abstract: need to be re-written. It is not expressed properly.
I read whole manuscript and found several faults (grammatical and spelling, scientific names, etc). I strongly recommend to revise the MS thoroughly to correct all the errors before resubmission.
Introduction need to be improved.
L71 use “and” instead of “&”
L77 delete “(A. thaliana)”
L77 what do you mean by “the exploitation tool”
There must be consistency for the titles in whole manuscript. Some place authors used lower case letters (ex. L72) and in some place used higher case letters (ex: L85)
L86-88 need to re-written “To elucidate the evolutionary relationships between the identified members of the PDCB gene 87 family in the cotton genome, a phylogenetic tree was constructed, through utilizing the PDCB 88 amino acid sequences from A. thaliana, following A. thaliana's grouping standards”
L108-113 this sentence need to be rewritten
L144-146 is highly confusing. Need to be précised.
L184-193 This paragraph need to rewritten

Figures need to be improved.

Experimental design

L98-102 I am surprised why authors went for finding the chromosomal localiazation of identified genes. Because, once you BLAST with cotton genome, you will find the BLAST hits on chromosomes with their physical positions in the initial step itself.

L105 why did authors used 2000bp upstream of start codon for cis-regulatory elements analysis?

RNA experiment need to be explained properly

L203-220 I am surprised, why authors just randomly picked 12 genes from the pool for qRT-PCR analysis? Since, expression analysis is so sensitive, anybody can get certain level of differential expression across multiple tissues over different time points. However, the thing is, we need to check the expression of particular gene at particular time point for specific trait. Random selection and conducting qRT-PCR is not the scientific way for doing the expression analysis. Just to improve this part of study, I will recommend to do the expression analysis for remaining genes, that will make sense in some way.

L112 what is the criteria for selecting “(5 DPA), 10 DPA, 15 DPA, and 20 DPA” as the time points for RNA experiment? Don’t you think, 5DPA is too early? Again, why authors did not discuss the qRT-PCR results in abstract?

Validity of the findings

L75 which is the cotton reference genome used for extracting PDCB genes? If it is the TM-1, then mention the version of genome.
L175-182 I recommend to follow the standard naming pattern for the chromosomes. Since, upland cotton is tetraploid, authors need to mention the chromosomes with their sub-genome (ex. A1, A2, etc) not as chromosome 1, 2, etc. This must be done in whole manuscript.

L194-202 since authors goal is to characterize the PDCB genes and finding their relevance in fiber length, it is unnecessary to talk about the CREs associated with abiotic stress. I recommend to search or look for the CREs involved in cell wall development or involved in fiber development. That will make more sense, rather than looking for abiotic stress.

Additional comments

I suggest to cite the most interesting recent article published in cotton quality traits.
• Gudi S*, Pavan M, Alagappan P, Raigar OP, Halladakeri P, Gowda RSR, Kumar P, Singh G, Tamta M, Susmitha S, Amandeep, and Saini DK (2024) Fashion meets science: how advanced breeding approaches could revolutionize the textile industry. Crit. Rev. Biotechnol. doi: 10.1080/07388551.2024.2314309

Figures need to be improved.

Reviewer 3 ·

Basic reporting

This manuscript reported a genome-wide identification of PDCB in G. hirsutum. After the identification, a candidate has been selected based in silico and gene expression analyses. VIGS experiment was done to suggest the involvement of the candidate gene, GhPDCB9, in fiber development and elongation. The manuscript is well-written but there are many points that are ambiguous. English can be further improved, better to get proofread by a fluent speaker. Use bigger characters in all figures since it is difficult to read in the current version. Avoid using those yellow characters in Fig 3, difficult to read. Figure legend are too short and did not provide complete information. All figures + legend must be able to stand alone. Put a scale bar in Fig 7a and c. No legend for sub-figures, a,b,c..

Experimental design

Methods section is needed to be improved.
Line 110, I am confused with 'samples were collected from fibers and three tissues (stem, leaf, and fibers)'. So how were the fiber samples different? two types fibers? Please revise sample details.

Line 123-124, use the prime symbol for 5' (prime) not apostrophe. Why the authors only use one reference gene beta-actin. Better to use at least two reference gene in RT-qPCR experiments.

Replace qRT-PCR with RT-qPCR which stands for reverse transcription-quantitative PCR.

Line 205, 'randomly selected 12 genes from the same family' I do not understand this sentence. Which family did the authors focused? why?

Why did the authors choose GhPDCB9 for VIGS? why not GhPDCB22?

Line 127: Virus Induced Gene Silencing: What is the control condition and how did the authors obtain the construct(s) for this control? For RNAi-induced silencing of gene expression, how long was the specific DNA fragment (bases XX–XX bp) located at the 3′ region of the cDNA which was inserted into plasmid? Please provide more details about the VIGS.

I did not find any information on the number of replicate used.

Statistical analysis must be performed and re-analyses; e.g., Fig 6 no stat data. Fig 7 b (no stat) and d (re-analyse, not appropriate).

Validity of the findings

The identification and in silico analysis are fine. The authors also provided a possible function of GhPDCB9 by VIGS experiment.

Additional comments

The Identification of PDCB genes in upland cotton: from blast search using A. thaliana as the query, how many matches did you obtain (paralogs and/or isomers)? Did you filter some out? How did you confirm the matches to be a member of the PDCB gene family? Any conserved domain? Please provide more details.

You identified 20 motifs. How about the information on functions of these motifs? Any data available in the literature?

Based on the promoter analyses, what are the predicted functions of the PDCB genes?

Please provide more discussion about the tissue-specific expression of the PDCB genes? any correlation with the predicted functions based on the promoter analyses (previous point)?

Discussion section needs more information regarding the main findings of this study, such as the interpretation of the phylogeny, conserved motifs, and the cis-regulatory elements found in the promoter regions. Please strengthen your Discussion.

---

## Round 0.2 · Major Revisions

Dear Authors

The manuscript cannot be accepted for publication in its current form. The manuscript needs substantial revision to meet the standards of the journal. The authors are invited to revise the paper considering all the suggestions made by reviewers. Please note that requested changes are required for publication.
With Thanks

·

Basic reporting

• Please merge the sentences in lines 58 and 59 as they are short.
• Line 93: Arabidopsis thaliana needs to be italicized.
• Line 95: replace 1000 with 1,000.
• Line 117: Gossypium hirsutum L. needs to be italicized.
• Line 200: G. hirsutum needs to be italicized.
• Line 231: The authors mentioned a “previous QTL research” but a citation from this work is missing.
• Line 254: the word “later” is misspelled.
• Figure 6: The graphs can be better visualized with improved picture quality: kindly upload a higher-quality of picture.

Experimental design

No comment

Validity of the findings

No comment

·

Basic reporting

Authors addressed all my queries

Experimental design

Authors carried/conducted experiment in proper way

Validity of the findings

Authors validated few CGs through qRT-PCR

Additional comments

Authors addressed all my queries

Reviewer 3 ·

Basic reporting

Although the authors have significantly improved the manuscript, there are still some points to be clarified/revised. The authors must carefully read their manuscript, otherwise it would be difficult to accept if another revision round is required.

Abstract
Abstract can be more concised. Do not repeat content, e.g. line 20 “identify PDCB family members within the Gossypium hirsutum genome” and line 21 “the PDCB gene was isolated from the Gossypium hirsutum genome”
Line 17, the full name of PDCB must be provided before the abbreviated form can be used.
Line 22, use italic font for G. hirsutum. Also use G. hirsutum, do not use Gossypium hirsutum after the first use (line 20). Please check this through the whole manuscript.
Line 24, ‘uniformity’? not ‘similarity’? better to confirm.

M&M
Line 119, please be more specific, stems, leaves, flowers at which stages? Young leaves? Mature leaves?
The authors did not mention anything on the number of biological replicates used in their experiments. How were the statistical analysis performed must be mentioned in this section, not just only in figure legends.

Fig 7, the authors did not use ‘*’, why put it there in the figure legend? Fig 7b and 7d, do you need to put the black squares to represent strain? You only used the single strain? Why the gene names are there? You only have one gene in this figure? Please check all figures since this is not a raw figure for publication.

Experimental design

no comment

Validity of the findings

no comment

Additional comments

no comment

---

## Round 0.3 · Minor Revisions

Dear Authors

The manuscript still needs a minor revision before it can be published. The authors are invited to revise the paper considering all the suggestions made by the reviewer.

With Thanks

Reviewer 3 ·

Basic reporting

The authors have revised their manuscript according to the comments but there are places to be corrected.

Line 17, after Plasmodesmata callose-binding protein, add '(PDCB)'. This must be mentioned as it is the first use.

Line 39, 'PDCB' not in italic font.

My previous comment (comment 5) on stages of stems, leaves, .... have not been address yet in the text. This must be provided. 'young stem' 'old leaves' young leave'..... please be more specific.

Many figures in the Review Materials are in low resolution. Please check the quality of the figures required by the journal.

Experimental design

no comment

Validity of the findings

no comment

---

## Round 0.4 · accepted · Accept

Dear Authors,

I am pleased to inform you that the manuscript has improved after the last revision and can be accepted for publication.
Congratulations on accepting your manuscript, and thank you for your interest in submitting your work to PeerJ.
With Thanks